# Effects of computerised clinical decision support systems (CDSS) on nursing and allied health professional performance and patient outcomes: a systematic review of experimental and observational studies

Teumzghi F Mebrahtu [ID] ,[1] Sarah Skyrme,[2] Rebecca Randell [ID] ,[3,4] Anne-Maree Keenan,[2] Karen Bloor [ID] ,[5] Huiqin Yang,[2] Deirdre Andre,[6] Alison Ledward,[2] Henry King,[2] Carl Thompson[2]

For numbered affiliations see end of article.

**Correspondence to**
Teumzghi F Mebrahtu;
t.f.mebrahtu@leeds.ac.uk

## ABSTRACT

**Objective** Computerised clinical decision support systems (CDSS) are an increasingly important part of nurse and allied health professional (AHP) roles in delivering healthcare. The impact of these technologies on these health professionals' performance and patient outcomes has not been systematically reviewed. We aimed to conduct a systematic review to investigate this.

**Materials and methods** The following bibliographic databases and grey literature sources were searched by an experienced Information Professional for published and unpublished research from inception to February 2021 without language restrictions: MEDLINE (Ovid), Embase Classic+Embase (Ovid), PsycINFO (Ovid), HMIC (Ovid), AMED (Allied and Complementary Medicine) (Ovid), CINAHL (EBSCO), Cochrane Central Register of Controlled Trials (Wiley), Cochrane Database of Systematic Reviews (Wiley), Social Sciences Citation Index Expanded (Clarivate), ProQuest Dissertations & Theses Abstracts & Index, ProQuest ASSIA (Applied Social Science Index and Abstract), Clinical Trials.gov, WHO International Clinical Trials Registry (ICTRP), Health Services Research Projects in Progress (HSRProj), OpenClinical(www.OpenClinical. org), OpenGrey (www.opengrey.eu), Health.IT.gov, Agency for Healthcare Research and Quality (www.ahrq.gov). Any comparative research studies comparing CDSS with usual care were eligible for inclusion.

**Results** A total of 36 106 non-duplicate records were identified. Of 35 included studies: 28 were randomised trials, three controlled-before-and-after studies, three interrupted-time-series and one non-randomised trial. There were ~1318 health professionals and ~67 595 patient participants in the studies. Most studies focused on nurse decision-makers (71%) or paramedics (5.7%). CDSS as a standalone Personal Computer/LAPTOP-technology was a feature of 88.7% of the studies; only 8.6% of the studies involved 'smart' mobile/handheld-technology.

**Discussion** CDSS impacted 38% of the outcome measures used positively. Care processes were better in 47% of the measures adopted; examples included,

### Strengths and limitations of this study

► The review is based on a comprehensive literature search.
► This is the first systematic review of clinical decision support systems influence on nursing and allied health professional (AHP) performance and outcomes.
► AHPs are under-represented, with a primary focus on paramedics and physiotherapists.
► The number of studies, service users/patients and health professionals involved was sizeable, but outcomes were too heterogeneous to aggregate.
► The overall quality of comparative research represented by the included studies was poor.

nurses' adherence to hand disinfection guidance, insulin dosing, on-time blood sampling and documenting care. Patient care outcomes in 40.7% of indicators were better; examples included, lower numbers of falls and pressure ulcers, better glycaemic control, screening of malnutrition and obesity and triaging appropriateness.

**Conclusion** CDSS may have a positive impact on selected aspects of nurses' and AHPs' performance and care outcomes. However, comparative research is generally low quality, with a wide range of heterogeneous outcomes. After more than 13 years of synthesised research into CDSS in healthcare professions other than medicine, the need for better quality evaluative research remains as pressing.

## INTRODUCTION

Nurses and allied health professionals' (AHPs') judgements and decisions commit financial, human and technical resources to care in health systems.[1] To support decision-making and underpin new roles and ways of delivering services, such as nurse-led

primary care,[1] computerised clinical decision support systems (CDSS) have been developed to tailor evidence-based advice provided to clinicians at the point of decision-making.

CDSS can improve professional performance by making the basis for decisions explicit; widening available information, encouraging more consistent decisions and thus reducing unwarranted variation in processes and patient outcomes.[2 3] Negatively, CDSS could encourage a focus on unimportant problems, hinder care delivery and contribute to a widening of (digital) inequalities.[4–6]

Reviews focusing mainly on doctors suggest CDSS effects on performance and outcomes are inconsistent,[7] but improved care processes[8 9] and reduced morbidity[8] and mortality[10] are possible. These reviews, however, often neglect the multidisciplinary nature of healthcare delivery and the decisions involved.

Previously synthesised studies of nurses' use of CDSS suggest only limited impact on performance and health outcomes.[11] Digital technology and research evidence have both developed significantly since this review was undertaken. In this review, we aim to examine the impact of CDSS on nurses' and allied health professionals' (AHPs) performance and patient outcomes.

## REVIEW METHODS
Following best practice principles,[12 13] we undertook a systematic review of research into CDSS targeting nurse and AHP decision-makers. The protocol was registered with PROSPERO[14] (number: CRD42019147773).

### Literature searching
Initial searches were conducted in November 2019 and updated on 12 February 2021. Searches were not restricted by language. See online supplemental table 1 for search terms.

We searched: MEDLINE(Ovid), Embase Classic+Embase (Ovid), PsycINFO (Ovid), Health Management Information Consortium (HMIC) (Ovid), AMED (Allied and Complementary Medicine) (Ovid), CINAHL, Cochrane Central Register of Controlled Trials (Wiley, Cochrane Database of Systematic Reviews (Wiley), Social Sciences Citation Index Expanded (Clarivate), ProQuest Dissertations and Theses Abstracts and Index, ProQuest ASSIA (Applied Social Science Index and Abstract), Clinical Trials.gov, WHO International Clinical Trials Registry (ICTRP), Health Services Research Projects in Progress (HSRProj), OpenClinical(www.OpenClinical.org), OpenGrey (www.opengrey.eu), Health.IT.gov, Agency for Healthcare Research and Quality (www.ahrq.gov).

### Study inclusion and exclusion
All titles and abstracts were imported into a reference management database (EndNote) and duplicates removed. Covidence review production toolkit (www.covidence.org) was used to manage screening, data extraction and organising of the review and ensure efficient production. After removing duplicate titles and abstracts, seven reviewers (A-MK, CT, HY, HK RR, SS and TFM) independently screened all titles and abstracts. TFM first-screened titles and abstracts for all studies, the other six authors then second-screened 16.7% of the studies each. Records with decision disagreements were revisited by two authors (TFM and CT) and resolved by consensus, a third reviewer (RR) was available for further disagreements although none occurred. Two reviewers (CT and TFM) independently assessed study relevance using Cochrane Collaboration's Effective Practice and Organisation of Care (EPOC) criteria;[15] and, conducted full-text screening. Any disagreements were resolved by consensus.

Comparative studies (randomised controlled trials (RCTs), non-randomised trials, controlled before–after (CBA) studies, interrupted time series (ITS) studies and repeated measures studies) comparing CDSS against usual care (ie, clinical decision-making unsupported by CDSS) were eligible for inclusion.

### Participants
Studies that evaluate the effects of CDSS used by *nurses (including midwives)* and AHPs and report professional performance and patient outcomes were eligible for inclusion.

### Interventions
The eligible intervention in this review was the use of *any form of CDSS to aid clinical decision making*.

### Comparator
The comparator was *usual care*, defined as *clinical practice where clinical decision making is unsupported by CDSS*.

### Outcomes
Our primary outcome was *adherence of nurses and AHPs to evidence-based recommendations*. Secondary outcomes were *diagnostic accuracy, time to reach judgement, adverse events, health professional satisfaction and system and/or implementation costs and benefits*.

### Data extraction
Data on study characteristics and outcomes were independently extracted by two reviewers (CT and TFM) using the EPOC standard data collection form.[16]

### Quality assessment
Study quality and risk of bias was assessed independently by CT and TFM using Cochrane Handbook for Systematic Reviews of Interventions[17] and EPOC guidelines.[18]

Each potential source of bias was judged as high, low or unclear, and an overall 'risk of bias' classification (high, moderate or low) assigned to each included study.[17] Studies with low risk of bias in all domains, or where bias was unlikely to fundamentally alter results, were treated as low risk. Studies with bias risk in at least one domain, or where bias might alter conclusions, were treated as unclear. Studies with a high risk of bias in at least one

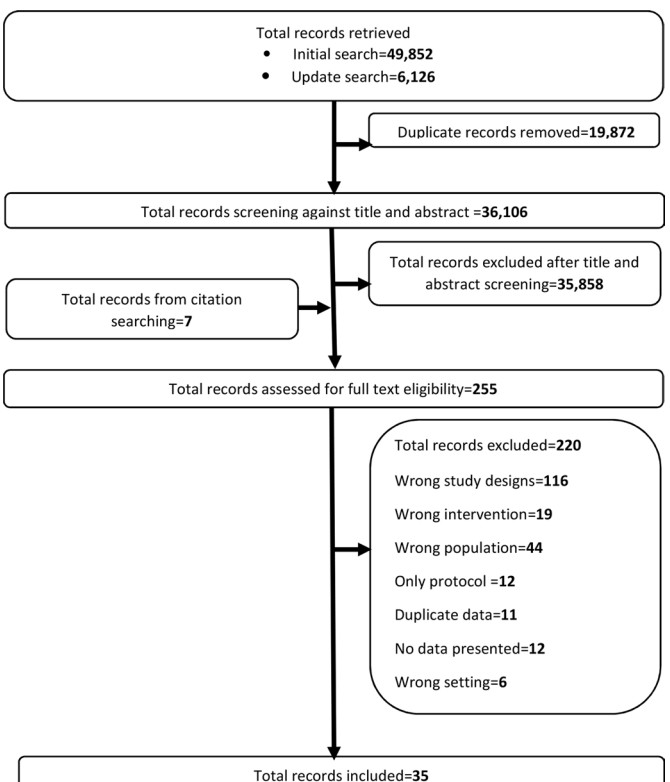

**Figure 1** PRISMA flow chart of study selection process. PRISMA, Preferred Reporting Items for Systematic Reviews and Meta-Analyses.

domain, or with a serious bias likely to reduce the certainty of conclusions, were considered high risk.

### Data synthesis

Findings were synthesised narratively, regardless of statistical analysis in the primary study. Studies were grouped by (i) similarity in focus or CDSS-type (knowledge based or machine learning), (ii) health professionals targeted, (iii) patient group, (iv) outcomes reported and (v) study design.

If not reported, we calculated absolute risks from the primary research. Risk differences and 95% CIs were then calculated from these. Because the CDSS, participants and underlying research questions were so heterogeneous no meta-analysis was undertaken.[19]

### RESULTS
#### Evidence quantity

From 36106 non-duplicate records identified, 35858 records were excluded after title and abstract screening. Seven records were identified through forward citation searching. Full-text screening was undertaken on 255 records which led to 220 more records being excluded. Thirty-five studies were included in the review.[20–51] Figure 1 illustrates study selection.

#### Study descriptions

The 35 included studies comprised 28 RCTs (80%), three CBA studies (8.6%), three ITS (8.6%) and one

non-randomised trial (2.8%). Thirty-two studies (91.4%) were peer-reviewed journal articles and three (8.6%) were PhD theses. The public sector funded 74.3% of studies; industry, 5.7%; 17.1% failed to declare funding and 2.9% were unfunded. Most studies were published after 2010 (n=29, 82.9%) with just two studies during 1997–1999 and 14 (40.0%) in 2000–2010. Sixteen studies (45.6%) were published after the last significant systematic review on CDSS for nurses' performance and health outcomes.[11] Circa 1318 health professionals and 67595 patients were study participants, mainly in hospital-based studies (57.1%). Primary care accounted for 17.1% and nursing homes 11.4% of studies. Western health systems provided the dominant context: US (28.6%); UK (20.0%), Netherlands (17.2%), Czech Republic and Norway (5.7%) each; with single study representation (2.8%) from Belgium, Brazil, China, Ghana, Norway, Sweden, Turkey and one multicentre (Austria, Czech Republic, and UK) report. See table 1.

Only one study (of 35) reported explicit theory to guide implementation of the CDSS. Almost a third (28%) published their study protocol—none of which discussed theory-influenced implementation.

Nurses made up the target for the CDSS *and* control groups in 25 (71.4%) studies; paramedics in two (5.7%) studies. Five studies (14.3%) compared nurses in the intervention (CDSS) group with physicians in the control. Two studies (5.7%) recruited a combination of nurses and physiotherapists for CDSS and control groups. Thirty-one studies (88.7%) used a standalone (physically, even when integrated in an electronic health record) computer-based CDSS; three (8.6%) used handheld/mobile-based technologies and just one study (0.2%) used a web-based CDSS. CDSS were mostly designed with a single function in mind (eg, disease diagnosis), but some addressed multiple parts of clinical pathways (eg, disease diagnosis *and* disease management).

#### Quality of identified evidence

Except for three RCTs scored as 'Unclear', all studies were at 'high' overall risk of bias. On average, RCTs scored 'Low' risk of bias in five of nine domains; CBA studies were lower, with four domains; non-randomised studies scored 'low' for a single domain. The three ITS studies were 'Low' risk of bias in six (of seven) domains. Evidence quality did not change over time (see online supplemental table 2).

#### Effects of intervention

Most studies reported more than two outcomes from a total of 124 individual outcomes reported (115 distinct types of measured outcomes). There were five distinct outcome groups:
► Care processes: aspects of patient data collection and management, and the process of patient management.
► Care outcomes: patient health outcomes (eg, fall and pressure ulcer prevention rate).

**Table 1**  Baseline characteristics of included studies

| Author and year | Country | Design | Setting | Study duration | Healthcare professionals (HP) | Outcomes |
|---|---|---|---|---|---|---|
| Beeckman et al[20] 2013 | Belgium | RCT | Nursing homes | 5 months | Nurses and physios | Risk of pressure ulcers; HP knowledge and attitude |
| Bennet et al[21] 2016 | UK | ITS | Emergency department, district general hospital | 1 year | Nurses | Triage prioritisation; pain assessment and management; management of neutropenic sepsis |
| Blaha et al[22] 2009 | Czech Republic | RCT | ICU postelective cardiac surgery university hospital | 48 hours | Nurses | Intensive care glycaemic control/diabetes |
| Byrne[23] 2005 | USA | CBA | Nursing homes | 33 months | Nurses | Falls and pressure ulcer reduction (assessment and prevention) |
| Canbolat et al[24] 2019 | Turkey | Non-RT | ICU university general hospital | 22 months | Nurses (and physicians) | ICU glycaemic control |
| Cavalcanti et al[25] 2009 | Brazil | RCT | ICU general hospital | 19 months | Nurses | ICU glycaemic control |
| Cleveringa et al[26] 2008 | Netherlands | RCT | Primary care practices | 1 year | Nurses (and physicians) | Management and prevention of diabetes (and CV risk factors) |
| Cleveringa et al[27] 2010 | Netherlands | RCT | Primary care practices | 1 year | Nurses | Management and prevention of diabetes (and CV risk factors) |
| Cortez[28] 2014 | USA | RCT | Academic medical centre oncology clinics | 11 weeks | Nurses | Management of cancer symptoms |
| Dalaba[29] 2015 | Ghana | CBA | Primary care health centres | 2 years | Nurses | Maternal care |
| Dowding et al[30] 2012 | USA | ITS | General hospitals | 6 years | Nurses | Risk assessment, falls and pressure ulcer prevention |
| Duclos et al[31] 2015 | France | RCT | Paediatric wards in a university hospital | 2 years | Dieticians | Nutritional care in malnourished children |
| Dumont et al[32] 2012 | USA | RCT | ICU wards in a regional referral hospital | 4 months | Nurses | Glycaemic control |
| Dykes et al[51] 2009 | USA | RCT | Urban hospitals | 6 months | Nurses | Fall prevention |
| Dykes et al[54] 2020 | USA | ITS | Academic medical centres | 42 months | Nurses | Fall prevention |
| Fitzmaurice et al[33] 2000 | UK | RCT | Primary care/general practice | 1 year | Nurses | Oral anticoagulation care |
| Forberg et al[34] 2016 | Sweden | RCT | Paediatric university hospital | 3 months | Nurses | Management of peripheral venous catheters in paediatrics |
| Fossum et al[35] 2011 | Norway | CBA | Nursing homes | 2 years | Nurses | Preventative behaviours and management of nutrition |
| Geurts et al[36] 2017 | Netherlands | RCT | University paediatric hospital | 2 years | Nurses | Management of (re)hydration in children |
| Hovorka et al[37] 2007 | Czech Republic | RCT | Cardiac Surgery, University Hospital | 48 hours | Nurses | Glycaemic control |
| Kroth et al[38] 2006 | USA | RCT | University Hospital | 9 months | Nurses | Body temperature assessment |
| Lattimer et al[39] 1998 | UK | RCT | Primary care practices | 1 year | Nurses and physicians | Emergency call assessment |
| Lattimer et al[40] 2000 | UK | RCT | Primary care practices | 1 year | Nurses and physicians | Cost analysis of emergency call assessments |
| Lee et al[41] 2009 | USA | RCT | School of Nursing (University) | 8 months | Nurses | Obesity management |
| Lv et al[53] 2019 | China | RCT | Community healthcare centres | 1 year | Nurses | Chronic asthma management |

Continued

**Table 1** Continued

| Author and year | Country | Design | Setting | Study duration | Healthcare professionals (HP) | Outcomes |
|---|---|---|---|---|---|---|
| Mann et al[42] 2011 | USA | RCT | Surgical Military hospital ICU | 6 days | Nurses | Glycaemic control in burn intensive care patients |
| McDonald et al[43] 2017 | USA | RCT | Nursing care homes | 2 months | Nurses | Management of chronic medical condition |
| Paulson et al[52] 2020 | Norway | RCT | University hospital | 10 months | Nurses | Management of malnutrition |
| Plank et al[44] 2006 | Mixed (Austria, Czech Republic, UK) | RCT | University hospitals | 48 hours | Nurses | Glycaemic control |
| Rood et al[45] 2005 | Netherlands | RCT | Surgical ICU in a teaching hospital | 10 weeks | Nurses | Glycaemic control |
| Roukema et al[46] 2008 | Netherlands | RCT | Children's Hospital | 27 months | Nurses | Management of children with fever without apparent source |
| Sassen et al[47] 2014 | Netherlands | RCT | University research centre | 17 months | Nurses and physios | Professionals' behaviour |
| Snooks et al[48] 2014 | UK | RCT | Emergency ambulance services | 1 year | Paramedics | Assessment and management of falls |
| Vadher et al[49] 1997 | UK | RCT | Cardiovascular medicine, general hospital | | A nurse and Trainee doctors | Oral anticoagulant control |
| Wells[50] 2013 | UK | RCT | Emergency ambulance services | 1 year | Paramedics | Emergency fall assessment and management |

CBA, controlled before and after; ICU, intensive care unit; ITS, interrupted time-series; RCT, randomised controlled trials.

▶ Health professionals' knowledge, beliefs and behaviours: outcomes that relate to the health professionals themselves (eg, changed attitude and perception due to CDSS use).

▶ Adverse events: safety issues that could arise due to the use of CDSS (eg, morbidity).

▶ Economic costs and consequences: outcomes that relate to direct costs, savings, or cost-effectiveness of CDSS.

### Care process

CDSS was better than usual care for 16 of 34 (47.0%) care process outcomes. Care delivery was worse (n=5, 14.7%) or no different for 13 (38.2%) processes. See online supplemental table 3.

### Adherence to guidelines

The four RCTs reporting nurses' adherence to guidelines examined 10 outcomes.[32 34 45 49] Only one trial reported baseline and follow-up data for both arms,[34] CDSS users had better adherence to hand disinfection guidelines (risk difference=6.7%; 95% CI: 4.9% to 8.5%); but were less likely to follow guidelines on disposable glove use (risk difference=−1.4%; 95% CI: −2.2 to −0.5%) and daily inspections of Peripheral Venous Catheters (risk difference=−5.2%; 95% CI: −7.2 to −3.3%).

Two trials[32 45] showed nurses using CDSS had better compliance with guidelines on insulin dosing (risk difference=22%; 95% CI: 19% to 25%) and on-time blood sampling (risk difference=4.7%; 95% CI: 2.0% to 7.4%). They deviated less from protocols (mean score difference out of 10=−2.6; 95% CI: −4.5 to −0.71) and concurred more with recommended insulin doses (than trainee doctors).[49]

### Patient assessment, diagnosis and treatment practices

Five RCTs[31 36 38 46 50] and one ITS[21] reported 18 indicators of patient assessment and treatment quality. Pain assessment quality (pain score use and appropriateness of choices) of emergency department patients improved by 62.7% (95% CI: 59.6% to 65.8%) and investigation of inpatient paediatric malnutrition aetiology was 21.2% higher (95% CI: 15.9% to 26.5%) with CDSS. However, optimal IV antibiotics administration for sepsis was lower reduced by 5.9% (95% CI: −8.3 to −3.5). Laboratory tests (electrolytes level acid–base balance test) and nutrition supplements (oral Rehydration Solution and intravenous rehydration) were no more likely to be ordered for paediatric inpatients by CDSS-enabled nurses.

There were marginally fewer wrongly recorded temperatures in hospital inpatients among CDSS-enabled nurses (risk difference=−0.8%, 95% CI: −0.9 to −0.6). Vital signs recording in patients attended by paramedics were also not significantly different.

### Documenting care

One ITS and a randomised trial reported five documentation-focused indicators.[30 52] Falls (risk ratio=1.4, 95% CI: 0.03 to 73.7) and hospital acquired pressure ulcer risk assessments (risk ratio=9.1, 95% CI: 1.95 to 42.5) were higher with CDSS. As was nutritional

 

care planning, food and fluid intake recording and treatment by nurses.[52]

### Referrals

Paramedics using CDSS were more likely to refer patients to a community falls than send them to the emergency department (risk difference=4.7%, 95% CI: 1.1. to 8.3).[48]

### Patient care outcomes

CDSS improved patient care outcomes in 22 of 54 (40.7%) indicators and worsened them for one outcome indicator (2.0%). See online supplemental table 4.

### Blood glucose control

Six RCTs[22 25 26 37 42 44] and one non-randomised trial[24] reported 19 indicators of glycaemic control, but only two reported baseline *and* follow-up values.[22 26] Blood glucose levels were better managed by ICU nurses using CDSS (mean=−2.2, SD=1.12) compared with paper-based *Mathias* (mean=−1.2, SD=0.66) and *Bath* (mean=−1.5, SD=0.78) protocols.[22] Glycated haemoglobin (A1C)<7%, systolic blood pressure <140 and total cholesterol <4.5 mmol/L were higher by 4.6% (95% CI: 2.7 to 6.5), 10.2% (95% CI: 7.9 to 12.5) and 3.7% (95% CI: 1.2 to 6.2), respectively, in patients receiving care from CDSS-enabled nurses compared.

Trials reporting only follow-up data suggest better blood glucose control by CDSS-using nurses across a range of indicators: proportion in target range (risk difference=32.9%; 95% CI: 20.0 to 46.0), occasions within the target glycaemic range (80–110 mg/dL) (risk difference=33.0%, 95% CI: 20.5 to 45.4), occasions over the target glycaemic range (>110 mg/dL) (risk difference=−31.0%, 95% CI: −43.7 to −18.2) and improvement of glycaemic control for 48 hours (risk difference=40.0%, 95% CI: 27.4 to 52.6)

### Blood coagulation management

One RCT reported three indicators of blood coagulation management in primary care.[33] Nurses using CDSS had significantly more tests in range (risk difference=4.0%, 95% CI: 0.4 to 7.6) than doctors *without* CDSS. However, the improvement from baseline was lower among nurses (risk difference=−1.9% (95% CI: −3.1 to −0.7), 'International Normalised Ratio (INR) Results within Range Point Prevalence' were not significantly different between the two groups and again, nurses using CDSS improved less than physicians without CDSS (risk difference=−2.6%, 95% CI: −5.3 to −0.1). There was no significant difference between groups in 'Time Spent within INR Target Range' (risk difference=7.0%, 95% CI: −0.7 to 14.7).

### Antenatal and peripartum care

The CBA study examining antenatal and peripartum care in community settings[29] suggested CDSS-using midwives reduced delivery complications (per 1000 attendances) compared with usual care (risk difference=2.4%, 95% CI: 1.1 to 3.7).

### Managing patients with chronic comorbid diseases

Two RCTs examined three indicators of successfully managing patients with complex chronic multimorbid health conditions in care homes,[43] and with asthma[53] showed no significant differences between CDSS users and non-users for emergency room usage, hospitalisation and complexity of medication regimens.

### Obesity screening

The RCT examining outpatient obesity screening by trainee nurses found CDSS-users had more 'encounters with obesity-related diagnosis' (risk difference=10.3%, 95% CI: 8.0 to 12.5) and fewer 'encounters with missed obesity-related missed diagnosis' (risk difference=41.0%, 95% CI: 48.8 to 35.0) than trainee nurses without CDSS.[41]

### Fall and pressure ulcer prevention and management

Two RCTs,[20 51] two CBA studies[23 35] and two ITS[30 54] focused on fall or pressure ulcer prevention and management. In a single trial,[20] pressure ulcer prevalence decreased more during the CDSS-enabled follow-up period (risk difference=−6.3%, 95% CI: −10.2 to −2.4), a result which was reversed in one of the CBA studies (risk difference=4.2%, 95% CI: 0.2 to 8.2).[35] The other CBA studies revealed no significant differences between CDSS using and non-using nurses trying to prevent falls and pressure ulcers.[23] In the ITS study, fall rate (risk ratio=0.91, 95% CI: 0.75 to 1.12) and hospital acquired pressure ulcer occurrence (risk ratio=0.47, 95% CI: 0.25 to 0.85) were significantly lower with CDSS.[30]

### Triage

Three RCTs[39 40 48] and one ITS study[21] evaluated CDSS impact on triage judgements. Health professionals using CDSS made fewer calls to general practitioners (GP) for telephone advice (risk difference=−34.2%, 95% CI: −36.0 to −33.0), had fewer patients visited at home by duty GPs (risk difference=−5.5%, 95% CI: −6.9 to −4.2) and fewer hospital admissions within 3 days (risk difference=−0.98%, 95% CI: −1.8 to −0.2) of the judgement. There were no differences in, 'patients left at scene without conveyance to emergency department' (risk difference=5.2%, 95% CI: −1.7 to 12.1). The ITS study reported the proportion of *correct (sic)* triage prioritisation judgements was higher among CDSS-users (risk difference=24.7%; 95% CI: 18.8 to 30.6).

### Quality of life and patients' satisfaction

Two RCTs examined CDSS impact on quality of life and patient satisfaction.[27 48] Patients in CDSS-using groups gained more life years (average difference in years=0.14, 95% CI: −0.12 to 0.40), more healthy years (average difference in years=0.04, 95% CI: −0.07 to 0.14) but reported lower quality of life and satisfaction. None of these differences were statistically significant.

### Health professionals' knowledge, beliefs, and behaviour

CDSS effects on knowledge, beliefs and behaviours of health professionals[20 28 32 47] were the focus of four RCTs

using 12 indicators. CDSS increased 'Positive knowledge change' (risk difference=6.5%; 95% CI: 0.8 to 13.2), 'positive attitude change' (risk difference=12.7%, 95% CI: 5.9 to 19.5), 'research utilisation' (risk difference=9%; 95% CI: 3.3 to 14.7), nurses' satisfaction (difference in satisfaction out of 10=3.6, 95% CI: 2.4 to 4.8) and perceived deviations from protocols (mean difference out of 10=−4.7, 95% CI: −6.1 to −3.3). Conversely, there was no significant impact on behaviours, intentions, perceived behavioural control, subjective and moral norms, barriers and research utilisation of CDSS-using nurses and physiotherapists (online supplemental table 5).

### Adverse events

CDSS are not risk free, and three RCTs[27 33 48] used four indicators to examine adverse events. Cardiovascular events in patients with diabetes (risk difference=−11.0%, 95% CI: −18.0 to −4.0) and deaths in primary care patients (risk difference=−5.7%, 95% CI: −10.1 to −1.7) were lower in CDSS-using groups of professionals. Serious adverse reactions in primary care patients and deaths in patients recently fallen and attended by paramedics were no less likely (online supplemental table 6).

### Economic costs and consequences

Four RCTs[27 36 40 48] used 20 indicators to report economic costs and consequences of CDSS. Costs of managing cardiovascular disease were lower in CDSS users (cost difference=−€587.00, 95% CI: −880.00 to −294.00). Diabetes care cost more (cost difference=€326.00, 95% CI: 315.00 to 318.00); took longer per care task ('mean length of job cycle time' difference in minutes=8.9; 95% CI: 2.3 to 15.3) to generate an additional quality adjusted life-year (QALY) costing €38 243.00 (online supplemental table 7).

## DISCUSSION

### Summary of main results

Our systematic review suggests that CDSS may improve some aspects of nurses' and AHPs' performance and care outcomes. Thirty-eight percent (38%) of indicators were better. Of 35 included studies, 26 (74.3%) reported CDSS-influenced care as better than care without CDSS on at least one outcome. In contrast, eight studies (22.8%) showed no significant difference between CDSS and usual care, with seven studies suggesting CDSS were less effective than usual care for at least one outcome.

### Care processes

Processes of care were better if CDSS was in use in almost half the studies, 16 of 34 (47%); a headline that masks a very wide range of absolute improvement: from 0.7% to 62.7%. Hand disinfection protocol adherence, insulin dosing, blood sampling at the right time and documented care were all better in CDSS users. This should be contrasted with the five (16.1%) outcomes where CDSS provided no advantages over usual care. Both sets

of findings are mitigated further by the considerable uncertainty in trying to estimate a holistic picture: the effects in 13 care process indicators (41.9%) were not estimable; either because studies lacked power (lower than minimum acceptable of 80%) to detect a difference in the comparison groups, or appropriate confidence intervals were not reported or could not be calculated from information published.

### Patient care outcomes

CDSS was associated with significantly better patient care outcomes across a broad range of 22 of 54 (40.7%) indicators (absolute difference between 4.6% and 42.9%). Just one indicator (1.8%) suggested no significant difference. Nurses using CDSS had better blood glucose control in emergency care patients (in five out of seven studies involved) and nurses and physiotherapists using CDSS were associated with better fall risk and pressure ulcer management. Triage was improved in nurses using CDSS in emergency call centres and paramedics faced with 'emergency falls' in older patients.

### Health professionals' knowledge, beliefs, and behaviour

Improved knowledge, beliefs and behaviour occurred in three of 12 indicators (25%). Nurse and physiotherapist CDSS-users had more knowledge and better attitudes compared with non-users. Compared with usual care, nurses utilised more research, were more satisfied at work, and perceived a greater need to follow protocols if they used CDSS.

### Adverse events

CDSS generated fewer adverse events across two of four indicators (50%). CDSS-using nurses had fewer cardiovascular events and reported deaths in their primary care patients compare to similar patients seen by doctors not using CDSS.

### Economic costs and consequences

CDSS did not significantly increase costs, or save money. Costs per QALY was €38 243.00 in one study—higher than the widely accepted willingness-to-pay threshold of €20 000 per QALY[27] and the UK *de facto* threshold of £30 000 per QALY to be considered cost-effective by the National Institute for Health and Care Excellence.[55]

### Comparison with other studies or reviews

Only one previous review has examined the effects of CDSS on nursing performance and patient outcomes.[11] Twenty new primary studies have been published since this review; but inconsistent outcomes and weaknesses in study designs and methods remain. Given the importance of implementation in effectiveness, it was noteworthy that most studies lacked a theoretical foundation for the implementation of CDSS. Similarly, many studies did not report using guidelines for designing, conducting/evaluating and reporting CDSS-use. Of 35 included studies, just one used an explicit implementation model/theory

at design stage.[20] None of the studies discussed their findings with reference to implementation science/theory.

In their review of 100 trials—principally with doctors—Garg *et al*[7] reported improved performance in 64% and better patient outcomes in 13% of studies. Our results suggest greater improvement may be possible for nursing work in particular (47% of process indicators and 41% of outcomes). Garg *et al*[7] transformed improvement into a binary (yes/no) indicator and did not quantify the outcome improvements—making the clinical significance of improvements hard to ascertain.

Bright *et al*[8] reviewed RCTs of CDSS with a range of health professional decision-makers (doctors, nurses and AHPs). They reported improvements in processes of care (OR=1.55, 95% CI: 1.38 to 1.74) and morbidity (RR=0.88, 95% CI: 0.80 to 0.96), but no impact on mortality (OR=0.79, 95% CI: 0.54 to 1.15) or safety/adverse events (RR=1.01, 95% CI: 0.90 to 1.14). However, outcomes measured were too heterogeneous for meta-analysis. The criteria for comparison groups were relaxed; the 'intervention' sometimes included paper-based decision support and alternative CDSS systems were used as a comparator in some studies. Our review required there to be an indication for the use of CDSS and a comparator that ruled out CDSS-use as part of 'usual care'. While we found improvements are *possible* from CDSS, comparison with Bright *et al*'s findings would be unreliable.

Moja and colleagues' review of 18 RCTs[10] (including nurses and AHPs alongside doctors) found no significant difference in CDSS-attributable mortality (RR=0.96, 95% CI: 0.85 to 1.08) but lower morbidity (RR=0.82, 95% CI: 0.68 to 0.99). While mortality and morbidity findings are similar to ours, their use of CDSS in the primary study comparator groups, again makes comparisons unreliable.

A recent review of 115 trials of CDSS, with a mix of health professionals, reported process improvements of the order of 5.8% (95% CI: 4.0% to 7.6%) with CDSS.[9] As with Bright *et al*, the 'comparator' criteria were unclear and outcome measures too heterogeneous for meta-analysis. Studies with more than two comparators were treated as different trials, meaning double counting and multiple comparisons (p-hacking) could not be ruled out, confounding comparisons with our findings.

### Strengths and limitations

Our review, while based on a comprehensive literature search, is a function of that literature. Consequently, we have highlighted primarily the impact of CDSS on nurses rather than AHPs. With the exception of paramedics and physiotherapists, other AHPs are poorly represented.

Evidence quality was poor and has not improved significantly since 2009. While the number of studies (35), service users/patients (~67 000) and health professionals (~1318) involved were sizeable, outcomes were too heterogeneous for aggregation. Inconsistencies in the effects of CDSS on target health professionals' performance and patient outcomes remain unresolved.

Moreover, although we have used a comprehensive list of databases in our search, the possibility of missing studies due to search terms cannot be ruled-out.

## CONCLUSIONS

CDSS can benefit nurse and (some) AHP delivered performance and patient outcomes. CDSS can improve adherence to guidelines and enhance patient care. Triaging of emergency patients, glycaemic control and screening of malnutrition and obesity all represent appropriate targets for CDSS. These conclusions require cautious interpretation: they are based on mainly low-quality studies, with heterogeneous outcomes and indicators.

To improve the quality of studies and consistency of outcomes, future research should satisfy two key requirements. First, system designers and evaluators should consider appropriate implementation theory/models (examples include Normalisation Process Theory[56] and the NASSS framework)[57] given the planned technology and associated work to encourage sustained adoption. Second, study reporting is varied, poor quality and lacking essential detail for implementation; guidelines for conducting and reporting CDSS should be a feature of the publication of findings. This would make synthesis easier and more informative. Guidelines for CDSS reporting in general already exist, it is difficult to conceive why they cannot be applied to nursing and AHP-focused CDSS.[58 59]

**Author affiliations**
[1]School of Computing, University of Leeds, Leeds, UK
[2]School of Healthcare, University of Leeds, Leeds, UK
[3]Faculty of Health Studies, University of Bradford, Bradford, UK
[4]Wolfson Centre for Applied Health Research, Bradford, UK
[5]Department of Health Sciences, University of York, York, UK
[6]Library Services, University of Leeds, Leeds, UK

**Contributors** A-MK, AL, CT, DA, HY, KB and RR contributed to conception of the review. DA conducted online database searches. A-MK, CT, HK, HY, RR, SS and TFM contributed to titles and abstracts screening. CT and TFM contributed to full-text screening, quality assessment and data extraction. TFM analysed and summarised data as well as produced the first draft of the manuscript. All authors have been involved in revising the work for important intellectual content and have approved the final version for publication. TFM had full access to all of the data in the study and take responsibility for the integrity of the data and the accuracy of the data analysis.

**Funding** The review is funded by the National Institute for Health Research Health Services Delivery and Research programme (award number: NIHR127926). The views expressed in this publication are those of the authors and not necessarily those of the NHS, the NIHR or the Department of Health.

**Competing interests** None declared.

**Patient consent for publication** Not required.

**Ethics approval** This study does not involve human participants.

**Provenance and peer review** Not commissioned; externally peer reviewed.

**Data availability statement** All data relevant to the study are included in the article or uploaded as supplementary information. Data used in the review are all available in this article and supplementary materials.

**ORCID iDs**
Teumzghi F Mebrahtu http://orcid.org/0000-0003-4821-2304
Rebecca Randell http://orcid.org/0000-0002-5856-4912
Karen Bloor http://orcid.org/0000-0003-4852-9854

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
