## [Reviewer comments · BMJ Open]

ARTICLE DETAILS

TITLE (PROVISIONAL)	Effects of computerised clinical decision support systems (CDSS) on nursing and allied health professional performance and patient outcomes: A systematic review
AUTHORS	Mebrahtu, Teumzghi; Skyrme, Sarah; Randell, Rebecca; Keenan, Anne-Maree; Bloor, Karen; Yang, Huiqin; Andre, Deirdre; Ledward, Alison; King, Henry; Thompson, Carl

VERSION 1 – REVIEW

REVIEWER	Tiase, Victoria University of Utah, College of Nursing
REVIEW RETURNED	05-Sep-2021

GENERAL COMMENTS	Thank you for this contribution. The methods are clear and well written. I have a few minor comments to improve this submission. 1. Please add a bit more detail to the procedures for the abstract/full text review as well as the extraction. What software was used? Was this all done on paper? How were disagreements handled with six reviewers?2. I only see one search strategy, but multiple databases were used. Could you please include all search strategies as part of the appendix?3. There are a number of limitations to this work. Could you please include the limitations of the search terms used and the databases selected? Thoughts on the use of grey literature?4. In each section, please be consistent with decimal point places. They are varied throughout the document.5. On pg18/line48, I'm unfamiliar with the word 'noughties' - should that be nineties? Thanks again.
--

REVIEWER	Vassilacopoulos, George University of Pireaus, Digital Systems
REVIEW RETURNED	07-Sep-2021

GENERAL COMMENTS	This paper reports on the impact of digital technology for decision support on healthcare professionals and patient outcomes by conducting a systematic review on this based on various databases, including MEDLINE, EMBASE and CINAHL. The results support to a significant level the positive impact hypothesis on selected aspects of nurses' and AHPs' performance and care outcomes. This is a quite comprehensive review of the scientific literature, making good use of the data collected in a descriptive, non-analytical, form on the various parameters considered. The paper is
---

	well presented with regard to the (non-analytical) descriptive method used and to the thorough discussion and literature cited for those interested in the particular field. Therefore, I would recommend that the paper is accepted for publication in its present form.
--	---

REVIEWER	Liebe, Jan-David Osnabrück University of Applied Sciences, WiSo
REVIEW RETURNED	08-Sep-2021

GENERAL COMMENTS	Dear authors, thank you very much for the exciting study, which I basically consider to be very relevant, profound and well done. Below are my notes and suggestions.  • Relevance: Highly relevant, focusing on the one hand on the still inconsistent findings on process and outcome-related outcomes, and on the other hand on the multidisciplinary nature of health care and related decisions. • Objective and research questions: The objective of the review is clearly motivated. The research questions addressed could be specifically identified (as found on PROSPERO, i.e.: What are the effects on clinical practice, the experience of patients and professionals, and the performances and outcomes of computerized clinical decision support systems (CDSS) used by nurses and allied health professionals (AHPs)?) • Method: The approach follows established best practices and was logged under PROSPERO. The primary and secondary outcomes recorded appear to be sufficiently broad in scope and at the same time defined concretely enough for a viable search strategy, i.e., adherence to evidence-based recommendations, diagnostic accuracy, morbidity, mortality, incremental cost-effectiveness ratios, etc. • Results: The study description is detailed and based on EOPAC. The results are described along five major outcome groups. The groups could be briefly defined. • Discussion: The discussion is also well structured, but I would like to mention two points that miss the focus of the review a bit (and probably can't be taken into account anymore), but may fit well into the discussion. (1) In principle, it would be interesting to know whether the time to the onset of the effects was also reported and whether there are any indications regarding the sustainability of the effects (e.g. regarding outcome group 1: Are there indications that the processes "dilute" again after a certain time or that the processes revert to old patterns)? I am also interested in whether there is evidence of moderating, mediating and confounding factors. So do the studies report on facilitating and inhibiting factors in the successful implementation of CDSS? (2) I am also interested in whether there is evidence of moderating, mediating and confounding factors. In its current state, the review gives a good description of the expected effects of CDSS use for nursing / AHPs. At the same time, it comes out that these effects are not always proven, which is certainly not only due to the study design but also to the actual implementation in the care processes. A systematic review of corresponding evidence would be helpful.
--

VERSION 1 – AUTHOR RESPONSE

Reviewer: 1

Dr. Victoria Tiase, University of

Utah Comments to the Author:

Thank you for this contribution. The methods are clear and well written. I have a few minor comments to improve this submission.

COMMENT 1: Please add a bit more detail to the procedures for the abstract/full text review as well as the extraction. What software was used? Was this all done on paper? How were disagreements handled with six reviewers?

AUTHORS' RESPONSE: Thank you, we have now added a text to clarify this in the “study inclusion and exclusion” sub-section.

COMMENT 2: I only see one search strategy, but multiple databases were used. Could you please include all search strategies as part of the appendix?

AUTHORS' RESPONSE: Thank you. We have now added all the database-specific search strategies in the “supplementary file” now.

COMMENT 3: There are a number of limitations to this work. Could you please include the limitations of the search terms used and the databases selected? Thoughts on the use of grey literature?

AUTHORS' RESPONSE: Thank you. We have this statement (“Moreover, although we have used a comprehensive list of databases in our search, the possibility of missing out studies due to search terms can be ruled-out.”) in the limitations section. We would like to bring to the reviewer’s attention that we did in fact search for grey literature as part of our search strategy. There were no included “grey literature” studies and so discussion of their effect on findings would be somewhat superfluous.

COMMENT 4: In each section, please be consistent with decimal point places. They are varied throughout the document.

AUTHORS' RESPONSE: Thank you. We have revisited the results section and made sure the decimal point places are consistent.

COMMENT 5: On pg18/line48, I'm unfamiliar with the word 'noughties' - should that be nineties?

AUTHORS' RESPONSE: Thank you. We are referring to the period 2000-2009. We recognise that it was an inappropriate term and have replaced it with “2009” with no loss of meaning.

Reviewer: 2

Dr. George Vassilacopoulos, University of Pireaus

Comments to the Author:

Please see attached file

AUTHORS' RESPONSE: Thank you for the attached positive feedback.

Reviewer: 3

Dr. Jan-David Liebe, Osnabrück University of Applied

Sciences Comments to the Author:

Dear authors, thank you very much for the exciting study, which I basically consider to be very relevant, profound and well done. Below are my notes and suggestions.

COMMENT 1- Relevance: Highly relevant, focusing on the one hand on the still inconsistent findings on process and outcome-related outcomes, and on the other hand on the multidisciplinary nature of health care and related decisions.

AUTHORS' RESPONSE: Thank you for the positive feedback.

COMMENT 2-Objective and research questions: The objective of the review is clearly motivated. The research questions addressed could be specifically identified (as found on PROSPERO, i.e.: What are the effects on clinical practice, the experience of patients and professionals, and the performances and outcomes of computerized clinical decision support systems (CDSS) used by nurses and allied health professionals (AHPs)?)

AUTHORS' RESPONSE: Thank you, we have slightly modified our two statements of the "objective" section of the abstract so it is clearer now. We have also explicitly included a PICO (participants, Interventions, Comparator and Outcomes) section in the methods for clarity and consistency with our PROSPERO record.

COMMENT 3-Method: The approach follows established best practices and was logged under PROSPERO. The primary and secondary outcomes recorded appear to be sufficiently broad in scope and at the same time defined concretely enough for a viable search strategy, i.e., adherence to evidence-based recommendations, diagnostic accuracy, morbidity, mortality, incremental cost- effectiveness ratios, etc.

AUTHORS' RESPONSE: Thank you for the positive feedback.

COMMENT 4-Results: The study description is detailed and based on EOPAC. The results are described along five major outcome groups. The groups could be briefly defined.

AUTHORS' RESPONSE: Thank you, we have added brief descriptions for the groups—see effects of intervention sub-section in the results section.

COMMENT 5-Discussion: The discussion is also well structured, but I would like to mention two points that miss the focus of the review a bit (and probably can't be taken into account anymore), but may fit well into the discussion.

AUTHORS' RESPONSE: See our responses below.

COMMENT 5.1: In principle, it would be interesting to know whether the time to the onset of the effects was also reported and whether there are any indications regarding the sustainability of the effects (e.g. regarding outcome group 1: Are there indications that the processes "dilute" again after a certain time or that the processes revert to old patterns)?

AUTHORS' RESPONSE: A great couple of points; as we have stated in the results section, all included studies reported only one estimate for the post-intervention period. The authors of these studies also do not report the onset of the 'effect'. Hence, estimating the sustainability/"half-life" of any 'effects' from the published information is impossible for many studies.

COMMENT 5.2: I am also interested in whether there is evidence of moderating, mediating and confounding factors. In its current state, the review gives a good description of the expected effects of CDSS use for nursing / AHPs. At the same time, it comes out that these effects are not always proven, which is certainly not only due to the study design but also to the actual implementation in the care processes. A systematic review of corresponding evidence would be helpful.

AUTHORS' RESPONSE: Again, a good point but one which goes beyond the scope of this publication

– aimed at presenting the "headlines" of our review. As highlighted in our discussion, the studies we included in this review lacked/had not reported any theoretical basis of implementation and evaluation of the CDSS used; and, did not report on facilitating and inhibiting factors for implementation of CDSS. The consideration of moderating, mediating and confounding factors by the study authors is entirely absent. Therefore, although one can speculate that there could be factors that affect success of CDSS implementation, it was not evidenced in the studies included in this review. In a separate work in progress we are looking at whether factors known to influence CDSS adoption and effectiveness (in medically focussed evaluations and syntheses) apply to nursing and AHP focused systems. This is a complex piece of work, due in no small part to the

scant reporting of systems and implementation in extant study articles. We would hope to publish this as a separate publication at some point in the future in BMJ Open when the analysis is complete. If we do we will certainly link it to this publication (if successfully published) and we would hope that the combined effect of both analyses will go some way to addressing your well-articulated uncertainty.